# Care Coordination Needs of Families of Children with Down Syndrome: A Scoping Review to Inform Development of mHealth Applications for Families

**DOI:** 10.3390/children8070558

**Published:** 2021-06-29

**Authors:** Beth Skelton, Kathleen Knafl, Marcia Van Riper, Louise Fleming, Veronica Swallow

**Affiliations:** 1School of Nursing, University of North Carolina at Chapel Hill, Chapel Hill, NC 27599, USA; kknafl@email.unc.edu (K.K.); vanriper@email.unc.edu (M.V.R.); lkflemin@email.unc.edu (L.F.); 2Department of Nursing and Midwifery, Sheffield Hallam University, Sheffield S1 1WB, UK; v.swallow@shu.ac.uk

**Keywords:** down syndrome, chronic/long-term condition, intervention, health, child/children, family, mHealth, care coordination, health management, health information

## Abstract

Care coordination is a critical component of health management aimed at linking care providers and health-information-involved care management. Our intent in this scoping review was to identify care coordination needs of families of children with Down syndrome (DS) and the strategies they used to meet those needs, with the goal of contributing to the evidence base for developing interventions by using an mHealth application (mHealth apps) for these families. Using established guidelines for scoping reviews, we searched five databases, yielding 2149 articles. Following abstract and full-text review, we identified 38 articles meeting our inclusion criteria. Studies incorporated varied in regard to research designs, samples, measures, and analytic approaches, with only one testing an intervention by using mHealth apps. Across studies, data came from 4882 families. Common aspects of families’ care coordination needs included communication and information needs and utilization of healthcare resources. Additional themes were identified related to individual, family, and healthcare contextual factors. Authors also reported families’ recommendations for desirable characteristics of an mHealth apps that addressed the design of a personal health record, meeting age-specific information needs, and ensuring access to up-to-date information. These results will further the development of mHealth apps that are tailored to the needs of families with a child with DS.

## 1. Introduction

Care coordination is critical for the management of care for any child with a chronic condition, but particularly for those with a genetic condition that could span multiple body systems, such as children with Down syndrome (DS). Management of the care provided by members of the healthcare team from a variety of healthcare settings is called care coordination [1]. Care coordination is a central component in guidelines set forth by the American Academy of Pediatrics (AAP) policy on family-centered medical home and has been shown to improve family-centered outcomes [2]. Researchers have defined essential components to be included in successful implementation of care coordination in primary care; these include being family-centered and comprehensive in nature to meet both health and psychosocial needs [3]. As a part of care coordination, it is important to include all persons who may be involved in the management of care for a child. This includes community service providers, such as physical and speech therapists, as well as those involved in the management of individual educational plans (IEP) [4].

Care coordination is critical to ensuring adequate management of co-occurring conditions with DS. The American Academy of Pediatrics (AAP) along with the Down Syndrome Medical Interest Group (DSMIG), led a task force in developing care guidelines for providers to manage the care coordination needs of children with DS [5]. These guidelines encompass not only co-occurring conditions commonly associated with DS but also well-child care including developmental screenings and immunizations. However, research has shown that provider adherence to these guidelines can vary greatly from zero completion of screenings to completion of more than 75% of the recommendations, with only an estimated 9.8% of children with DS being up to date of recommendations for care [6,7]. Missing these recommended care considerations could lead to complications from co-occurring conditions, as well as gaps in care [7,8]. Further, it has been found that successful implementation of care coordination can increase completion of well-child care [9]. Care coordination can also influence caregivers’ perceptions of how well the care provided meets the needs of their families [2,10].

When thinking about care coordination, it is also critical to consider who is responsible for managing the care of a child with DS. For children with medical complexities, the burden of management of health information and care coordination often falls to the parent or caregiver [11]. There are also barriers from the healthcare providers in aiding families in implementing care coordination, such as lack of personnel, lack of communication skills, and lack of time [12]. While there have been excellent developments in health information management and communication through the use of electronic health records (EHR), because of the possible range of persons involved in care, much of the health information ends up outside of a single electronic health record system. Given these barriers, mHealth apps may be a way to fill this gap by supporting caregivers and families in managing their care coordination and health information management needs.

The World Health Organization defines mHealth as any “medical and public health practice supported by mobile devices, such as mobile phones, patient monitoring devices, personal digital assistants, and other wireless devices” [13] (p. 6). There are many potential uses of mHealth apps including communication needs between individuals and health services, health monitoring, and access to information [13]. Studies have shown that primary caregivers are open to the idea of using mHealth apps to support management of their child’s chronic condition and recognize that technology may facilitate improved information access and communication [14,15,16]. To begin the development of an intervention such as an mHealth app, it is first important to gain a full understanding of family and caregiver needs and preferences. This scoping review was designed to examine the extent and nature of research on caregivers’ role in care coordination by addressing the following questions: (1) What are the care coordination needs of primary caregivers of children with DS? (2) What strategies and resources do primary caregivers use to address their care coordination needs? These findings will help guide user-centered design of an mHealth app to support families of children with DS.

## 2. Methods

### 2.1. Protocol

The protocol for the search and analysis was developed by using guidelines outlined by Arksey and O’Malley (2005), as well as Preferred Reporting Items for Systematic Reviews and Meta-Analysis Protocols for Scoping Reviews [17,18].

### 2.2. Eligibility Criteria

Listed below are the inclusion and exclusion criteria developed for selection of articles to be included in our scoping review [17].

Inclusion criteria include the following:Articles appearing in peer reviewed journals;Research implementing any study methodology and design;Published in journals between January 2010 and January 2020;English language;Study population includes caregivers of children with DS birth to twenty-one years old;Article addresses health management of child with DS, including dental management;Article reports results reflecting perspectives of parents.

Exclusion criteria include the following:Not research;Age of child with DS not reported;Focused exclusively on educational management of a child with DS;Did not differentiate children with DS from other possible conditions included in the sample;Focused exclusively on the prenatal period or diagnosis experience;Review articles;Reporting only the providers’ perspective.

### 2.3. Information Sources

With guidance from a research librarian, a literature search was completed on 21 January 2020. Databases included in the search were CINAHL, Embase, ProQuest Health Management, PsycInfo, and PubMed. These databases were selected to capture journals from a range of healthcare disciplines, as well as research on health management. The final search results from each database were exported to Endnote, where duplicates were removed. Results were then exported to Covidence to complete both the title and abstract screening, as well as the full-text review (Veritas Health Solutions, Melbourne, Australia).

### 2.4. Search

Search terms were intentionally broad to cast a wide net to find all studies possibly addressing care coordination in the context of DS [17]. While each search was modified slightly to meet the constraints of the search tool, the PubMed (Medline) search string was as follows: [(mother* or father* or parent* or caregiv* or family* or families* or “family”[mesh] or “pediatric*” or “child”[mesh] or “child, preschool”[mesh] or “adolescent”[mesh] or “teen” or “teenager” or “child*” or “infant*” or “toddler”) AND (“care coordination” or “coordination” or “patient-centered care”[mesh] or “case management”[mesh] or “meaningful use”[mesh] or “case managers”[mesh] or “health communication”[mesh] or “needs” or “health care” or “healthcare”) AND (“Down syndrome”[mesh] or “down syndrome”) and “last 10 years”[PDat]]. In addition to database searches, reference lists of each article included in the review were hand checked to identify studies that were not captured within search strings.

### 2.5. Selection of Sources of Evidence

All studies were screened by two reviewers, using Covidence software at both the title and abstract screening, as well as the full-text review level (Veritas Health Solutions, Melbourne, Australia). The first two authors resolved disagreements for inclusion by discussion to reach final consensus.

As the full-text review was completed, it became evident that additional inclusion and exclusion criteria were required to successfully identify studies related to the research questions [18]. As part of the inclusion criteria, studies needed to have a focus on health management; however, this definition was further refined to include interaction with the healthcare system and providers.

### 2.6. Charting the Data

A structured template for charting the data was developed by the first two authors (BS and KK). Data categories included in the template were as follows: study purpose, study design, respondent, conceptual underpinnings of study, definition of care coordination, age range of children in sample, measures, use of technology, and findings related to care coordination and health-information management. Following the development of the extraction template, BS and KK independently charted the data from five studies and then met to discuss the completeness and clarity of the extraction template and make final revisions. Extractions were completed in Microsoft Word. As a check on the extraction process, KK checked every fifth extraction for accuracy and completeness. Only minor changes were needed, providing evidence of the quality of the first authors’ extractions.

### 2.7. Collating and Summarizing Data

To collate and summarize study findings, the Marshall definition of care coordination and the Family Management Style Framework (*FMSF*) structured the analysis [19]. Marshall (2019) defined care coordination as “[relying] on communication between primary care providers and specialty care services and access to and facilitation of services and support” and was used to analyze the data [20] (p.79). While the purpose statement gave a glimpse into how each study addressed care coordination, within this scoping review it was also necessary to review the studies’ findings to answer each of the research questions. While there were 28 articles in which the study purpose included an element of care coordination, all articles were included in this analysis, using both the purpose statement and study findings. The constant comparison method was used to develop themes across the purpose statements and results from each study [21]. Summaries of each of the themes is included in the results.

## 3. Results

The search of five databases yielded 2149 articles for review after duplicates were removed. Of those articles, 2147 came from database searches, and two articles came from reference-list screening. Following PRISMA guidelines (see Figure 1), screening of studies by title and abstract review excluded 2074, leaving 77 articles for full-text review. Thirty-eight articles from 37 studies were included in the final sample for extraction. Thirty-nine articles were eliminated after the full-text review. The reasons for exclusion are listed in the figure below (Figure 1).

### 3.1. Characteristics of Sources of Evidence

Data extracted from each study are summarized in Table A1 (Appendix A). Findings summarized in extraction included the study’s first author, country where data were collected, study purpose, study design, respondent, age of child with DS, measures, and technology use. These data were extracted to provide a broad overview of the type of studies investigating caregiver interactions with healthcare with regards to care coordination.

### 3.2. Study Design

The 38 articles in the final sample included multiple study designs. Articles were evenly distributed between qualitative and quantitative study designs. Fourteen studies were qualitative [22,23,24,25,26,27,28,29,30,31,32,33,34,35], eleven were quantitative [36,37,38,39,40,41,42,43,44,45,46], and five were mixed methods [20,47,48,49,50]. Investigators in six studies completed secondary analyses, four using data from the National Survey of Children with Special Health Care Needs (NS-CSHCN) [51,52,53,54], one completing a chart review [7], and one using the Intellectual Disability Exploring Answers (IDEA) database [55]. Only two studies were longitudinal in design [55,56]. Finally, one study employed a pre-/post-test quasi-experimental design [57].

### 3.3. Measures

There was a lack of consistency across studies in measures used to assess characteristics of care coordination. Outside of the secondary data analyses using the NS-CHCSN survey, only two studies used the same care coordination measure. Caregivers’ management of their child’s special healthcare needs with DS was assessed by using standardized tools such as the Family Experiences Survey (FES), Family Problem Solving and Communication (FPSC), Family Management Measure (FaMM), and Parenting Stress Index (PSI). Each of these measures was used in two studies included in the review [20,36,38,48,56,57]. Additionally, two studies used the Oral assessment in DS Questionnaire [37,44]. The Emotionality, Activity, Sociability Survey, Family Adaptability and Cohesion Evaluation Scale, the brief Family Assessment Measure (FAM), Family Support Scale (FSS), Family Coping Index (FCI), Family Environment Scale, Family Index of Regeneratively and Adaptation (FIRA-G), Parent-caregiver Perception Questionnaire (P-CPQ), Perceived Social Support Scale (PSSS), Family Assessment Device (FAD), Beck Depression Inventory (BDI), Index of Social Competence (ISC) and Family APGAR were each used in one study [36,38,40,43,56,57]. Nine studies used measures generated by the investigators to assess caregiver needs [22,39,42,43,45,46,47,49,50].

In the qualitative studies, investigators collected data through interviews or focus groups, using interview/discussion guides developed for the study. Twelve qualitative studies collected interview data [23,25,26,29,30,31,32,33,34,35,49,50]. Investigators in four studies collected data from focus groups [24,27,29,48].

### 3.4. Analysis

Several different analytic methods were employed to analyze qualitative data. In five studies, investigators analyzed interview data by using a grounded theory approach [25,26,31,32,49]. Krueger and colleagues were the (2019) only investigators to describe using triangulation for analysis [28], while van den Driessen Mareeuw (2019) was the only author to report using framework analysis [35]. Finally, investigators in six studies used content analysis [23,29,30,33,34,50]. Cartwright and Boath (2018) analyzed focus group findings by using a phenomenological approach [24], while the others completed a content analysis [27,29,48].

Descriptive statistics were reported for all eleven quantitative studies, as well as the five mixed-methods studies [20,36,37,38,39,40,41,42,43,44,45,46,47,48,49,50]. Twelve of the studies including quantitative data completed tests of association or correlation [20,37,38,40,41,42,43,44,45,48,49,50]. Frequency counts were employed by five studies as part of the analysis [20,39,44,45,47]. Regression modeling was used in six studies including investigations of family adaptation [36], functioning and social support [38], care transitions [41], uncertainty and communication [42], social competence [43], and attitudes towards mHealth app usage [46].

### 3.5. Respondents

3024 families of children with DS were included across 31 primary studies. Eliminated from this total were secondary data analyses and chart reviews. In five secondary analyses, investigators from four studies used data from the National Survey of Children with Special Healthcare Needs years 2005/2006 (1128 participants) or 2009/2010 (504 participants), and in one secondary analysis, the investigator used data from the Health Utilization Survey completed in Western Australia between 1997 and 2004, to which 121 families of children with DS responded [51,52,53,54,55]. Lastly, one investigator completed a chart review of 105 medical charts of children with DS [7]. 

When identified, the primary caregiver in 20 studies, was the mother. Five studies focused exclusively on mothers. Of the 3024 families, 1669 respondents identified as mothers. Only one study exclusively focused on fathers, with 93 respondents. Across studies, 416 respondents identified as fathers. Few studies identified other family member respondents. Bertoli et al. (2011) included 100 siblings of persons with DS [47], and Parrott et al. (2012) included 18 sibling respondents, as well as caregivers [42]. Three studies each had one respondent identifying as a grandmother [30,48,49]. The remaining studies did not specify which parent or caregiver completed the study. Only one study focused exclusively on caregiver dyads [57]. Several studies included caregiver dyads within the sample but did not analyze the data as dyads [27,32,34,37]. One study focused on caregiver-and-child dyads; however, health-related quality-of-life data were only collected from mothers and were linked to the child’s dental-health physical exam findings [40]. While the focus of this scoping review was families and caregivers of children with DS, several studies also included healthcare providers, with 72 provider respondents.

### 3.6. Age of Children with DS

Data were extracted regarding the age range of the children with DS. There was great variation in how studies reported age of the child with DS. Since the respondent in each of the studies reviewed was the caregiver and not the child, data on the child were often incomplete or not included. Four studies did not include the age range of the children with DS. Twenty-one studies reported the average age of the child with DS. The overall average age across these studies was 9.24 years. Thirteen studies reported only an age range and not an average age of the child with DS. While the focus of the scoping review was children and adolescents under the age of 21, some of the studies also included adults with DS. The age ranges reflect this inclusion ranging from birth to 61 years of age. In 6 articles from 5 studies, the sample included caregivers of both children and adults with DS.

### 3.7. Study Purpose

The purpose of study each article was examined. Although all studies addressed caregivers’ interactions and experiences with healthcare systems and providers for their child with DS, only two studies explicitly used the term “care coordination” in the study purpose [20,53]. Additionally, 26 articles contained elements of healthcare management included in care coordination in their purpose statements. These elements related to care coordination included knowledge of services [22,33,37,41], communication [30,42], access to care [44], information availability [24,26,34,52], engagement and advocacy [28,35,50,51], and utilization and availability of services [29,43,46,47,49,55,56,57]. There were three studies that addressed the AAP guidelines for care of a child with DS [7,39,45]. The other ten studies contained elements in the purpose that could be related to care coordination but was not immediately apparent from the details included in the purpose statement and were found in the study results.

## 4. Research Question One

Research question one was designed to address what are the care coordination needs of primary caregivers of children with DS? A model was created from the themes developed from collating data collected from the purposes and results of each article (Figure 2). All 38 articles were analyzed for factors related to the care coordination needs reported by primary caregivers of children with DS as relates to research question 1. Themes are ordered by the frequency with which they occurred within each topic area.

## 5. Communication

Sixteen studies included in the scoping review reported results related to communication between caregivers and providers [20,23,24,26,27,29,30,31,34,36,42,48,49,50,51,53]. Caregivers’ perceptions of the quality of communication were linked to both positive [20,36,49,50,51,57] and negative [23,24,26,27,29,34,53] outcomes for families and children in both quantitative and qualitative studies. In four studies investigators focused on caregivers’ beliefs about the importance of communication rather than their assessment of the quality of their communication with professionals [30,31,42,48]. These findings, both negative or positive in nature, have the potential to impact the quality of communication and thus care coordination.

Shared decision making was a common theme that caregivers expressed as promoting effective and positive communication [30,49,50,51]. Communication was also found to be a protective factor for caregivers in developing resilience and family adaptation [36]. In Marshall et al. (2019), 88% of respondents expressed satisfaction with communication between healthcare providers, as well as satisfaction between providers and other professionals [20].

However, there were negative findings related to communication between caregivers and providers. Caregivers reported supportive communication to be rare and more often felt that providers did not understand family needs related to DS and frequently evoked negative emotions [24,27]. Caregivers expressed that there was the perception of information being withheld and caregivers did not feel supported by healthcare providers [23,26]. Almost half of caregivers (44.6%) were less than very satisfied with the communication between different providers involved in their child’s care [53]. Caregivers also described non-existent or scattered communication between providers when describing care coordination [29].

Four studies detailed how communication was a contributing factor in effective care coordination practices [30,31,42,48]. Hall et al. (2012) found the perception of open versus closed communication with providers to be influenced by the caregiver’s stress level, with stressed families describing more closed communication in focus groups [48]. Both Murphy et al. (2017) and Melvin et al. (2018) identified communication as a factor influencing caregiver perception of quality of life and self-management ability respectively [30,31]. Finally, Parrot et al. (2012) found that as caregiver uncertainty increased related to medical management, the caregivers’ desire for increased communication also increased [42]. These findings are important in understanding how communication may be both influencing and influenced by individual family member well-being.

Results related to communication also addressed caregivers’ perceptions of their communication with other parents of children with DS. Caregivers wanted to connect with other families of children with DS and learn from their experiences [20,23,34,57]. Caregivers reported that communication with other caregivers fostered the exchange of information, as well as creating a bond between families [23]. Caregivers described these experiences as having a positive impact on their ability to coordinate care.

## 6. Information

Eleven studies investigated the information needs of caregivers and families [20,23,24,26,29,30,32,34,46,50,57]. Included under this theme were study results addressing the quality and timing of information. Across studies, it was found that caregivers were motivated to acquire information regarding their child and DS [20,24,29,30,46,50,57]. However, there were several factors impacting the desire for information. Families wanted information to be from trustworthy sources [26] and were often frustrated by the quality and lack of information provided by healthcare providers [53]. Caregivers also reported feeling responsible for accessing reliable health management information [32]. Families were also concerned about providers not being up to date and knowledgeable in the management of health concerns associated with DS [23,24,30,34]. Caregivers were concerned that information presented in printed materials was often out of date and information on available resources was lacking [29].

Age and timing were also considerations in caregiver’s information needs. Families expressed a desire to have targeted information at times of transition or at specific age milestones [34,50]. However, families did not want to be overloaded by information [23,34]. In Choi et al.’s (2019) study of an mHealth app addressing family adaptation and therapeutic communication, researchers reported that families’ found information most helpful from birth to age 12 months [57]. Melvin et al. (2018) defined these critical information periods as early development ages (0 to 3 years) and school transitions and also specified that families should be offered information early in the child’s life to be able to develop realistic goals and expectations [30]. However, less than half of families reported receiving information about DS in the birth setting [20].

## 7. Utilization

In sixteen studies investigators linked elements of healthcare utilization to care coordination [7,20,29,31,39,41,43,44,45,46,47,50,51,53,54,55]. The most prominent feature of utilization was the role of the medical home. The medical home is a model of care defined as “holistic care of children and their families where each family has an ongoing relationship with a health care professional” [58] (p. 17A). Care coordination is a key component in the implementation of medical home policy [59]. Investigators found that children with DS were at significantly more risk for not having a medical home than other children with special healthcare needs [51]. Between 29.7 and 37.5% of caregivers of children with DS reported having a medical home compared 47.3% of families with other special healthcare needs reporting having a medical home [51,53]. While only about 1/3 of families reported having a medical home, 88% reported seeing a general healthcare provider in the past three years [43]. In studies for this review, children without a medical home were at greater risk for missed components of care management and failure to meet healthcare transitions [7,41,51,53,54]. Caregivers of children with DS without a medical home may be less prepared to aid their child with the transition from pediatric to adult healthcare because of a lack of support and encouragement from their healthcare provider [41]. Utilization was also investigated by three studies looking at the use of the AAP/DSMIG guidelines for the management of children with DS [7,39,45].

Investigators also examined the amount of time caregivers spent addressing their child’s special needs. Caregiver time is also an important consideration when investigating healthcare utilization. Examples of outcomes related to caregiver time included in these studies were issues such as travel time to appointments and missed days of work. However only one study investigated healthcare utilization related to travel times, reporting that 1/3 of the sample (*n* = 41) took more than an hour to reach a pathology center [46]. Caregivers reported missing on average 7 workdays annually for healthcare visits [46]. Phelps et al. (2012) reported that 30.2% of caregivers reported dedicating eleven or more hours per week to care coordination for their child with DS [53].

## 8. Experiences with Healthcare

While all of the studies included some connection to caregiver and family experiences with healthcare, there were thirteen articles that highlighted specific details with regards to these interactions [20,23,24,25,26,27,28,29,30,34,49,53,57]. Some of the results of caregiver experiences fell into other categories because they specifically addressed themes such as information, but findings from nine studies reported more general experiences with healthcare or experiences that did not fit into other categories described above [23,24,25,27,28,29,30,34,53].

These experiences related to caregivers’ general perceptions of the interaction with healthcare providers. Caregivers reported providers were helpful in providing information, however where providers fell short was understanding caregivers’ emotional responses to their child’s condition and special needs [34]. There were other experiences expressed across studies that were perceived as negative. For example, caregivers in several studies expressed dissatisfaction with healthcare providers who were perceived as insensitive and lacking support for behaviors such as breastfeeding [23,25,27]. Caregivers felt there was a lack of distinction in care needs for children with DS versus typically developing children [24]. In one study, less than half of the families reported feeling as though they are a partner in care with their child’s provider [53].

## 9. Individual Functioning

A second theme within individual and family factors addressed the individual functioning of a child with DS. Twelve studies included elements of individual functioning of children with DS [7,23,24,39,41,43,47,49,51,52,54,55]. Articles focused on various components of individual functioning including co-occurring conditions commonly associated with DS, impact on activities of daily living and demographic factors influencing functioning [7,39,41,43,47,52,54,55]. Two studies specifically addressed feeding concerns related to breastfeeding and feeding methods [23,24].

Several studies investigated the impact of functional status on the care received. Researchers reported there was variation in the specialty referrals for individual functional difficulties made by providers, leading to discrepancies in care received [49,54]. Children with higher functional impairments had lower odds of having a medical home, which studies have shown improves care coordination outcomes [51,59]. Children with DS were more likely to have delays in necessary therapies than children with other special healthcare needs [54].

## 10. Family Functioning

While family factors were addressed across thirteen studies [27,28,31,32,34,36,38,40,42,48,50,56,57], three factors of family functioning were identified as playing a role in care coordination, resiliency, advocacy, and uncertainty [28,42,48]. Families described care coordination across providers as one way they were able to advocate for their child, and this advocacy most common occurred in healthcare and school settings [28,32]. Caregivers’ perception of family resiliency also was a factor contributing family response to stressors [48]. Families who displayed more resiliency and less stress reported an increased perception of communication and support in a healthcare setting, important factors in care coordination [48]. Caregivers and families with increased uncertainty also expressed a strong desire to communicate [42].

## 11. Resources

Children with DS often require many resources to manage co-occurring conditions as a part of care coordination. These resources may be in the form of services such as therapies, as well as financial resources and transportation. Eleven studies included in our sample reported findings related to resources [22,26,27,29,31,32,36,41,46,47,49]. Most commonly financial constraints were reported as a barrier to receiving necessary care [22,26,27,32,49]. However, the lack of available social services and resources and how to access them was also reported as a concern for caregivers [20,29,41,47,49,54]. Investigators in one study reported that availability of community services was a positive indicator in family adaptation [36]. Caregivers reported frustration with resources from frequently having to change providers to maintain government supported services such as developmental therapies [31]. Support also varied greatly on the perceived level of the disability and the availability of the services [49].

## 12. Technology Use

The ultimate goal of this scoping review is to determine the content of an mHealth app that would best support caregivers of children with DS. Because of this goal, it was important to investigate if any of the studies incorporated the use of technology into management of care needs and how it may impact care. Choi and Van Riper (2019), studied family adaptation and DS, using an mHealth app to engage in therapeutic communication with a nurse. Investigators in a second study included in this review investigated caregivers’ attitudes and expectations for the use of an mHealth app [46]. Tozzi et al. (2015) found that almost all families connected to the internet either at home or work, and half of those families connected through a smartphone [46].

While technology was not included in the project aims, Melvin et al. (2018) found that caregivers expressed a desire to have an electronic way to organize information, as well as provide age-appropriate and research-based information to support their child’s communication [30]. Researchers also reported caregivers’ desire to have trusted information from online sources, particularly at times of transition such as the start of school, and the beginning of puberty [24,26]. Technology was also described as an important facilitator of communication between caregivers of children with DS through use of online forums, as well as platforms such as Facebook [23].

## 13. Dental Care

An unexpected finding of this scoping review were five studies exclusively addressing dental care [22,33,37,40,44]. While the focus was on dental care, many of the themes overlapped in terms of resources, information, and experiences with healthcare. Studies focused on dental care primarily reported frequency of visits, type of dental provider, use of anesthesia and knowledge related to the dental care of children with special needs [22,33,37,40,44]. Nqcobo et al. (2019) also reported on the frequency and treatment of dental caries in children with DS [40]. Rahim et al. (2014) found that children with DS received less dental care than typically developing children and none had received orthodontic care [44].

## 14. Research Question Two

A similar thematic analysis of the results was completed to address the second research question: What strategies and resources do primary caregivers use to address their care coordination needs? This question yielded fewer findings than the first question describing caregiver experiences with elements of care coordination. However, there were results that could inform future care coordination recommendations and be used in the development of interventions, using mHealth apps. Findings addressing the second research question fell within the themes of care coordination, family, and individual factors.

## 15. Care Coordination Factors

Caregivers’ recommendations addressing strategies and resources related to care coordination, included communication, information, and utilization. As reported in research question one, parents viewed shared decision-making as a strategy enhanced communication, particularly when initiated in the medical home [7,30,41,49,50,51]. This was also reflected in the need for care to be family centered [20,27,38]. Additionally, caregivers recommended strategies to maintain an organized, up to date health history as a way to minimize the time focused on updating the health history during appointments with providers [31]. This was particularly true for appointments related to child development [31]. Two studies recommended the use of mHealth apps to support time management and caregivers’ active participation in their child’s healthcare [46,57].

Because of the amount of information required for care coordination and health management, it was recommended that information be age-specific, as well as electronically available, to help manage the amount and timing of information [30,57]. Information was particularly important in transition planning [26,41,50]. Moreover, using knowledge gained on related protective family factors, such as communication, resources, and support, nurses can develop individual strategies for families related to adaptation and resiliency [36]. One study, by Thomas et al. (2011), made a recommendation related to resource use and utilization [55]. To maximize utilization, it was recommended by caregivers that providers have knowledge of health insurance coverage for specialty referrals [55].

## 16. Family and Individual Factors

Family factors also contributed to caregivers’ strategies for effective care coordination. Krueger et al. (2019) reported that caregivers recommended being assertive, as well as persistent, in their communication with healthcare providers to ensure receiving necessary services and information [28]. Families also need to develop good organization skills to manage care coordination [35].

## 17. Discussion

Findings from this scoping review support the conclusion that effective care coordination has the potential to increase family satisfaction and improve outcomes in managing care for a child with DS. These findings are consistent with study findings from a study of caregivers of adolescents with chronic conditions [10]. Care coordination has also shown the ability to reduce functional difficulties for children with special needs [60]. However, care coordination needs to be utilized in a manner that best supports individual family and child needs. Using the themes developed from the findings of this scoping review, we considered how these findings could be used to inform the development of an intervention, using mHealth apps for families. While fewer studies addressed research question two, there were findings from themes from both research questions that will aid in the development of an mHealth application.

The results of this scoping review point to reliable and up to date information as one of the most important caregivers needs with regards to care coordination for their child with DS. There are several findings that were identified from this scoping review that could be used to manage health information within an mHealth app to improve care coordination efforts. One area in which an mHealth apps could fill a gap is in the development of a personal health record (PHR). A PHR is designed to fill the information gap in the electronic health record by having persons or caregivers manage their own health information [61]. We know that children with DS have the potential to see many specialty providers in a variety of healthcare settings. A PHR may help organize required documentation of health behaviors for referrals [62]. For example, if there is a concern about obstructive sleep apnea, a PHR could serve as a journal for a caregiver to document sleep history [62]. The mHealth apps can be used to document information such as medications, provider information and lab results as part of a PHR [62]. Families expressed a desire to be able to keep health information in a way that is easy to bring with them between visits and an mHealth app is one intervention to fill this gap.

In conjunction with a PHR, other technologies are being developed to support caregivers of children with DS. Down Syndrome Clinic to You (DSC2U) is a web-based tool that will help caregivers identify referral needs for their child, using the AAP guidelines [63]. It may be possible for an mHealth app technology to work in conjunction with a website such as DSC2U for families to document information pertinent to referral appointments. The linking together of healthcare technologies increases caregivers’ ability to manage health information and care coordination.

However, one consideration to keep in mind when developing mHealth technologies is cost. Having mHealth apps be free and publicly available may help to increase parents’ access to health information needed to manage their child’s care. There is limited but positive evidence that mHealth apps are cost effective [64,65]. Examples of free mHealth apps focusing on the DS community include HealthSwap, focusing on the nutritional needs of persons with DS and mHealth apps targeting learning and communication needs such as the mHealth app Otsimo [66]. The DSC2U website mentioned above does carry a cost for use, but this cost can be offset by insurance companies [67]. However, there currently is no mHealth app addressing the care coordination needs of persons with DS. The global COVID-19 pandemic shed light on the importance of remote healthcare tools, and this is likely to continue into the future. As such, further research considering the cost and use of mHealth technologies is warranted.

Caregivers expressed the desire to have age specific health information as a part of coordinating care [34,50]. An mHealth app could allow for information to be communicated through pop-up reminders that could be tailored to a child’s age and the families’ specific needs. Pop-up reminders would be particularly relevant in infant and preschool years when there is an increase in specialty care visits and developmental changes [30]. Caregivers also wanted information at times of transition, such as the start of school and the beginning of puberty, and pop-ups could be used for this information [30]. This would also prevent the concern of receiving too much information all at once [34]. Pop-up information could also be used to create appointment reminders, reducing caregiver concern related to scheduling [53]. Finally, pop-up reminders could be used for caregivers to document information related to daily habits requiring documentation for co-occurring conditions as part of a PHR [62]. Pop-up reminders could serve many functions within a PHR and an mHealth app.

Additional key findings from this scoping review that can be used in development of an mHealth app include the need for up-to-date information and easy to access data [20,23,24,26,29,46,57]. These findings with regards to health information align with findings related to user centered design practices for developing mHealth apps. Desai et al. (2020) outlined six design preferences for mHealth apps [68]. Of these six design elements, using a table layout, a problem-based organization system, and linking content between different areas will help address these caregiver information needs [68]. Information within an mHealth app can be presented in a table format, as well as linking pertinent topics together for caregivers. The findings from this scoping review related to care coordination needs, as well as these design elements, will help develop an mHealth app that meets the needs of caregivers of children with DS.

## 18. Limitations

There were limitations to this scoping review. Articles were restricted to only those in the English language. However, the search terms for this scoping review were purposely broad in order to cast a wide net, creating great variety in the type of studies identified. There was a lack of continuity across studies in type of data, measures, and study design within the extraction. There was also inconsistent reporting on the age of the child with DS. Additional information on age would allow for additional analysis across studies based on the age of the child. It is possible that relevant articles may have been missed. This risk was minimized by seeking the input of a research librarian to design the search string. This scoping review was also limited by focusing exclusively on caregiver perceptions of care coordination. It was important to include healthcare providers’ input on care coordination and future reviews focusing on healthcare providers’ knowledge of care coordination, type of practice, and knowledge and ability to treat children with DS would also advance knowledge of strategies for ensuring optimal care coordination. Despite limitations, this scoping review provides valuable insights into families’ care coordination needs for a child with DS.

## Figures and Tables

**Figure 1 children-08-00558-f001:**
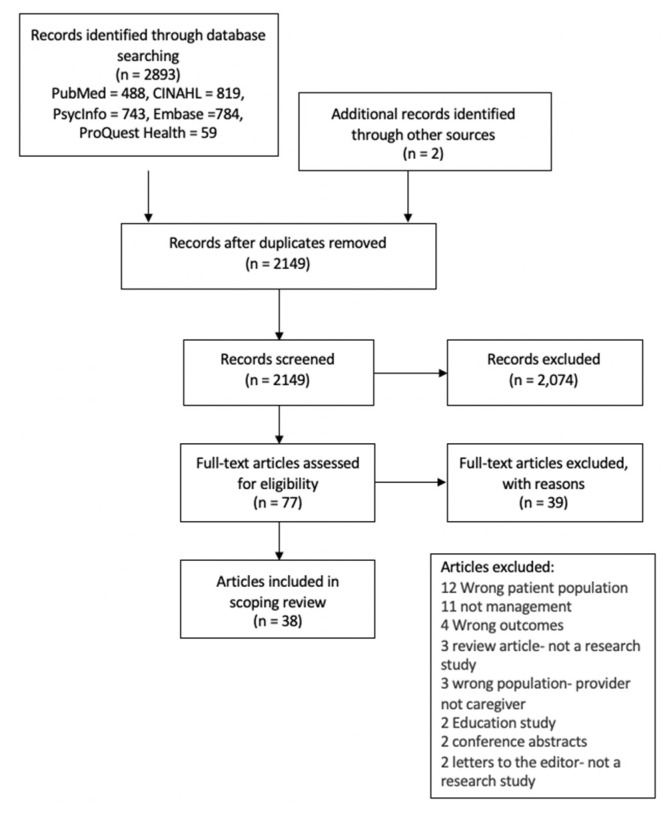
PRISMA diagram.

**Figure 2 children-08-00558-f002:**
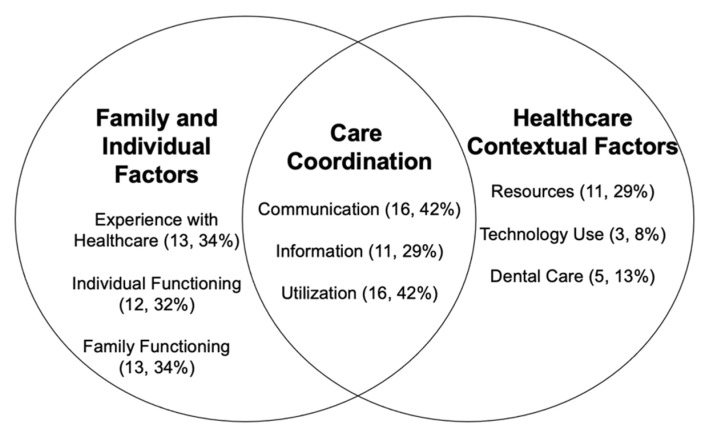
Themes related to care coordination. Number of studies addressing theme included in review, percentage of studies in review.

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
