# Peer review of "Care Coordination Needs of Families of Children with Down Syndrome: A Scoping Review to Inform Development of mHealth Applications for Families"

_children, 2021, doi:10.3390/children8070558_

Round 1
Reviewer 1 Report
This is very helpful research around the care coordination needs from a caregiver's perspective. Often care coordination is delivered from a clinical MD/RN/ SW perspective only and the perspective of family members is crucial. Additionally, this article may spur the development of creative use of patient electronic medical record portals to assist families caring for children with medical complexity. Overall a useful article for those who care for individuals with disabilities ... not just Down syndrome.
Reviewer 2 Report
Review of Care Coordination Needs of Families of Children with Down Syndrome: A Scoping Review to Inform Development of Mhealth Applications for These Families
The review article examined care coordination needs in families with Down syndrome. This review topic is relevant to current research in the field of intellectual and developmental disabilities and will be of interest to those involved with the treatment of children with DS. It also clearly outlines key components of care coordination and makes connections with current and needed development of mHealth apps. I commend the authors for taking this on, as the terminology in this area can be vague, and see that the results reflect a productive search even with these challenges. Overall, the article is well-written and clearly organized. The background information sets the reader up well to understand the article content. I would recommend this manuscript for publication, but also have comments to improve upon the text (see below).
- I suggest taking out “these” from the manuscript title. It seems to be implied which families you are referring to.
-A variety of articles are used before mHealth app (a, an, the, or no article). This is evident in the abstract but also persists throughout the manuscript. Please use the appropriate article and be consistent with the use throughout the text.
-Relatedly, it would be helpful to better define mHealth app(s). From other aspects of the manuscript it can be deduced that mHealth is a particular category of app but it would be helpful to be more explicit with the definition. It is helpful to give the WHO definition, but another sentence clarifying details as to who the apps are designed for and that this is a broad type of app would benefit the reader. It would also be helpful to know if care coordination is just one potential use for mHealth apps or if that is a primary objective/component.
-page 2, line 45 & 46: The sentence states, “in the development of IEP management”. I believe the authors either mean “in the development of IEP” or “in IEP management”. Please clarify.
-A small suggestion, on page 2 line 75: Would the first sentence be better suited later in the paragraph (perhaps moved before the sentence that starts “This scoping…”)? It may be beneficial to have this paragraph start with the definition information on mHealth, as the reader is looking for the information at that point in the text.
-page 3 line 136-7: Are the reference lists that were hand checked the reference lists in the articles found? Please be more specific here.
-page 7 line 284-285: There are two sentences that start with numbers, please write out the numbers at the beginning of the sentences.
-The “study purpose” (line 291) starts with “to address question one”, but it seems like the information in that paragraph is relevant to both study questions. Please adjust or clarify.
-At the top of page 8, would it be possible to add the research question into the introduction sentence? This was very helpful for the study question 2 section and would be nice to have here as well.
-page 9 line 370: the phrase “factors scaffolding the desire for information” is confusing. Does this mean “factors that impact access to information” or “factors required from information sources”? Please clarify.
-page 10 line 411-413: Are there any findings that could be summarized here from the studies mentioned?
-Please write out Down Syndrome Clinic to You when DSC2U is mentioned for the first time. Not all readers will be familiar with this tool.
-Consider mentioning the cost of DSC2U (and any other costs associated with mHealth apps mentioned). The development and use costs should be considered in the broader discussion of mHealth apps as a limitation or consideration for those interested in the distribution or use of these types of tools.
-Given the lack of articles related to study question 2, would it be appropriate to comment on the necessity for more research related to strategies caregivers use to address care coordination needs?
-page 14 line 612-614: This sentence includes two of the same phrase, “findings related to”, could the wording be varied to improve readability?
-Table a1 is thorough and helpful to the reader seeking additional details about an individual study.
